# RG: Out-of-Distribution Detection with Reactivate Gradnorm

## Abstract

Detecting out-of-distribution (OOD) data is critical to building reliable machine learning systems in the open world. Previous works mainly perform OOD detection in feature space or output space. Recently, researchers have achieved promising results using gradient information, which combines the information in both feature and output space for OOD detection. However, existing works still suffer from the problem of overconfidence. To address this problem, we propose a novel method called "Reactivate Gradnorm (RG)", which exploits the norm of the clipped feature vector and the energy in the output space for OOD detection. To verify the effectiveness of our method, we conduct experiments on four benchmark datasets. Experimental results demonstrate that our RG outperforms existing state-of-the-art approaches by 2.06% in average AUROC. Meanwhile, RG is easy to implement and does not require additional OOD data or fine-tuning process. We can realize OOD detection in only one forward pass of any pretrained model.

## 1 Introduction

In addition to the need for the accuracy of predictions, more and more attention has been paid to whether the model can make rejection identification when faced with completely unfamiliar samples. People want models that are not only accurate in their familiar data distribution but also aware of uncertainty outside the training distribution. This gives rise to the importance of out-of-distribution (OOD) detection, which determines whether an input is in-distribution (ID) or OOD. And OOD detection is widely used in fields with high safety requirements, such as medical diagnosis (Nair et al., 2020) and autonomous driving (Amini et al., 2018).

Deep neural networks can easily make overconfident predictions on OOD inputs, which increases the challenge to separate ID and OOD data Van den Oord et al. (2016); Chen et al. (2021). For instance, a model may wrongly but confidently classify an image of a crab into the clapping class, even though no crab-related concepts appear in the training set. Previous works focused on deriving OOD uncertainty measurements from the output space (Hendrycks & Gimpel, 2016; Liu et al., 2020) or feature space (Lee et al., 2018; Sun et al., 2022). A recent work (Huang et al., 2021) based on gradients has intrigued us. Actually, gradient information can often be decomposed into information from feature space and output space, which can be derived from the process of the BP algorithm. However, this method still has room for further improvement on OOD detection, which encourages us to utilize both output space and feature space information for better OOD detection.

In this paper, we perform OOD detection by jointly using information from feature space and output space. Formally, we propose Reactivate Gradnorm (RG), a simple and effective method to detect OOD by utilizing the inputs and outputs of the last layer of the neural networks. Specifically, RG directly uses the product of the 1-norm of the clipped input of the last layer of neural network and the logarithm of the exponential sum of the outputs (free energy) as the OOD scoring function. The reason for using the 1-norm of the hidden layer features is that the neurons will be activated for the ID sample. The motivation for cropping it comes from the fact that there will be a few OOD samples with strong features. Appropriate cropping can reduce the 1-norm of the features of the OOD samples without excessively affecting the 1-norm of the features of the ID samples. The energy information in the logits space is selected instead of the information in the probability space (like MSP(Hendrycks & Gimpel, 2016)) because there is information loss from the logits space to the probability space (the relative size information of the logits will be ignored by the softmax layer).

on the other hand, there are good theoretical and practical effects by using the energy as an OOD evaluation score.

Empirically, we have established excellent performance on the large-scale ImageNet benchmark. RG is vastly superior to previous use of energy after crop ReAct (Sun et al., 2021) 8.9% by AUROC, and our source of inspiration Grodnorm 5.86% by AUROC. Our method also achieves excellent performance compared to the MOS (Huang & Li, 2021) 2.06% by AUROC.

Our key results and contributions are summarized as follows:

- We propose RG, a simple and effective OOD uncertainty estimation method, which is label-agnostic (no label required), OOD agnostic (no outlier data required), train data agnostic (Only the pre-trained model is used and no fine-tuning or extra training).

- We conduct sufficient experiments on the combination of information from output space and input space to help us better understand the effectiveness of our OOD detection methods. RG promotes the average AUROC by 2.06% compared to the current best method under the same pre-trained model and dataset. Experiments show that using information from both feature space and output space has a gain for OOD detection.

- We perform a simple theoretical analysis of our method that using information from both feature space and output space at the same time helps to model the distribution of training data, which facilitates ood detection. And we unify several previous approaches under the equation 10 in a new framework.

## 2 BACKGROUND

In a supervised learning, we denote by $X = R^d$ the input space and $Y = \{1, 2, ..., C\}$ the output space. A neural network $f(x, \theta) = \{f_i(x, \theta)\}_{i=1}^C$ with the parameter $\theta$, we abbreviate it as $f(x)$, which is a mapping from $X$ to $R^C$. When given a dataset $D = \{(x_i, y_i)\}_{i=1}^n$, the supervised learning task is to minimize:

$$R(f) = E_{(x,y) \in D} l_{CE}(f(x), y) \tag{1}$$

where $l_{CE}$ usually used the cross-entropy loss:

$$l_{CE}(f(x), y) = -log \frac{e^{f_y(x)}}{\sum_{i=1}^c f_i(x)} \tag{2}$$

where $y$ is the ground-truth label.

**Problem statement** OOD detection can be formulated as a binary classification problem. The goal is to design a discriminator $G(x)$ which is a mapping from $X$ to $R$. Given a threshold $c$, we will decide a sample $x$ as an OOD sample if and only if $G(x) < c$. The design of the discriminator $G$ is often related to the neural network model $f(x, \theta)$, which will help the neural network model reject the recognition when $G(x) < c$. Typically, $c$ will be set to the fraction of 95% of In-distribution (ID) data that can to be identified as ID. The key challenge is to derive a scoring function $G(x)$ that captures OOD uncertainty. Previous OOD detection approaches primarily rely on the output or feature space for deriving OOD scores, and there has been some recent interest in utilizing gradient information for OOD detection. We will reveal that effective gradient-based OOD detection method is a method that combines the information from output space and feature space. And based on it, a more efficient method is proposed in the following section.

## 3 MOTIVATION AND METHOD

In this section, we will first describe the gradient-based OOD detection method and then analyze that the gradient-based OOD detection method is based on the synthesis of the information in the feature space and the information in the output space in section 3.1. The gradient-based OOD detection method inspired us to design an appropriate OOD score which is a combination of the norm of the clipped feature vector and the energy in the output space to achieve OOD detection in Section 3.2. In section 3.3, we unify several previous approaches under the equation 10 in a new framework.

### 3.1 GRADIENT-BASED OOD DETECTION

We start by introducing the loss function for backpropagation and then describe how to design the gradient norm for OOD uncertainty estimation.

We provide a perspective to revisit gradient-based OOD detection: the idea of the gradient-based OOD detection method stems from that *for a fully trained neural network, When we continue to feed the neural network with samples of the training set, then use the ground truth to calculate the loss and use the loss to backpropagation, the gradient of the neural network parameters will be small because the neural network is fully trained.*

However, when doing OOD detection, the ground-truth is missing. So we can not use the ground truth information to calculate prediction loss. A natural idea is to use a uniform distribution as a substitute for ground-truth and the gradient will be large for an ID sample. (Huang et al., 2021):

$$G(x) = ||\frac{\partial KL(u||softmax(f(x)))}{\partial w}||_1 \tag{3}$$

where $u = \{\frac{1}{C}\}_{i=1}^c$, $KL(u|softmax(f(x))) = -\frac{1}{C}\sum_{i=1}^c log\frac{e^{f_i(x)}}{\sum_{i=1}^c f_i(x)}$. $w$ is the parameters of the network.

Another way to replace the ground-truth is that we think the probability that the true label has a probability $p_y = \frac{e^{f_y(x)}}{\sum_{i=1}^c f_i(x)}$ to be class $y$. Then we can design the score as Igoe et al. (2022) mentioned:

$$G(x) = E_{y\sim p(x)}||\frac{\partial log p_y}{\partial w}||_1 \tag{4}$$

The probability that each sample has probability of $p_i$ belongs to the $i^{th}$ class, then the expectation of the gradient of the classification loss will be small for ID samples. Note that the negative log-likelihood is used when calculating the loss function, so for ID data, G(x) will also be larger than OOD data.

Similarly, to avoid the problem of missing ground truth, we can also use such a loss function $-\sum_{i=1}^C e^{f_i(x)}$ and design our own novel score as:

$$G(x) = ||\frac{\partial \sum_{i=1}^C e^{f_i(x)}}{\partial w}||_1 \tag{5}$$

Under some special settings: *only the gradient information of the last layer is used*, we will find out that equation 3 is actually:

$$G(x) = UV \tag{6}$$

where $V$ is the $L_1$ norm of the features of the last layer of neural network input, and $U = \sum_{i=1}^c |\frac{1}{C} - p_i|$. Meanwhile the equation 4 also has the form of equation 6, where $V$ is the $L_1$ norm of the features of the last layer of neural network input, and $U = 2\sum_{i=1}^c p_i(1-p_i)$. And the equation 5 also has the form of equation 6, where $V$ is the $L_1$ norm of the features of the last layer of neural network input, and $U = 2\sum_{i=1}^c e^{f_i(x)}$.

From the expression, these scores are still difficult to overcome the problem of overconfidence prediction of OOD samples. As the example we mentioned in the introduction, a model may wrongly but confidently classify an image of a crab into the clapping class, even though no crab-related concepts appear in the training set. Suppose the crab has a strong feature related to the recognition of the clapping class, which will result in a large 1-norm in the feature space. This means that there is room for improvement in both $U$ and $V$.

**Summarize** Some gradient-based OOD detection methods can be transformed into a combination of feature information and output information. But that's not surprising, because in the BP algorithm the gradient in the last layer is equal to the product of the error propagated and input value where the former only depends on the output space, the latter only depends on the feature space. Previous work inspired us to explore suitable $U$ and $V$ for OOD detection.

## 3.2 THE CHOICE OF U AND V

$U$ comes from the output space. In OOD detection, using the maximum softmax probability Hendrycks & Gimpel (2016) is a natural choice. However OOD samples may also have very confident predictions, so we use an energy score-based score as $U$:

$$U = Tlog\sum_{i=1}^{c} e^{f_i(x)/T} \qquad (7)$$

It has good theoretical and practical significance to use energy score as an indicator for OOD detection. It is also a very common strategy to estimate the certainty of predictions by summing the neural network output after activation, as in the method of estimating the certainty based on the Dirichlet distribution, using $softplus(\cdot) + 1$ as the activation and summing (Sensoy et al., 2018).

$V$ comes from the feature space. A common strategy is to model OOD data from a standard Gaussian distribution in feature space. For ood data, the $i^{th}$ element of features vector is $v_i = max(0, z_i)$, $z_i \sim N(0, 1)$. The score of ood is measured by $e^{-v_i^2}$. For ID data, use $1 - e^{-v_i^2}$ as measure. Based on the assumption that each feature in the feature space is independent, we should multiply all $V_i$ as V. But this will cause numerical instability, such as if one of the factors is 0 then the product is 0. So we use $V = \sum_i(1 - e^{-v_i^2})$. Actually we take an approximation of it: $V = \sum min(1, v_i)$. They are all based on the same idea: to avoid overconfident predictions of OOD samples with few strong features, the contribution of each feature to the overall score should be suppressed.

In general, the OOD score we use is:

$$G(x) = Tlog(\sum_i e^{f_i(x)/T}) \sum_i min(v_i, k) \qquad (8)$$

where $T$ is the temperature in the energy function, and default is 1. $k$ is the clipping threshold of each feature in the feature vector, and default is 1. In this paper, equation 8 is used as the OOD detection score by default.

## 3.3 ADDITION-BASED COMBINATION OF U AND V

Different from the combination method of multiplying U and V in the previous section, in this section we will use another method to combine U and V and provide a perspective that unifies the previous approaches.

To better explain this, let's look at the fully connected layer of the last layer of the neural network. Suppose for the sample x, the input of the last layer of the neural network is $v = \{v_j\}_{j=1}^{N}$. Suppose the joint probability that the feature $v$ belongs to class $i$ is $P(v, C_i) = e^{g(v)+w_i^t v+b_i}$, where $g(\cdot)$ is a mapping from $R^N$ to $R$, $w_i \in R^d$, $b_i \in R$. The choice of $g(\cdot)$ cannot be too arbitrary, as it will be constrained by probability normalization.

The probability that sample $x$ belongs to class $i$ is:

$$P(C_i|v) = \frac{P(v, C_i)}{\sum_{i=1}^{c} P(v, C_i)} = \frac{e^{g(v)+w_i^t v+b_i}}{\sum_{i=1}^{c} e^{g(v)+w_i^t v+b_i}} = \frac{e^{w_i^t v+b_i}}{\sum_{i=1}^{c} e^{w_i^t v+b_i}} \qquad (9)$$

This is exactly what the final fully connected layer of the neural network and the softmax layer are working. We will easily see the combination of the fully connected layer and the softmax layer ignores the relative size $g(v)$. This also implies that if the information of $v$ from feature space can be effectively used, the performance of OOD detection can be better than the information of the prob space alone, like MSP (Hendrycks & Gimpel, 2016).

Then,

$$logP(v) = log(\sum_{i=1}^{c} P(v, C_i)) = log(\sum_{i=1}^{c} e^{g(v)+w_i^t v+b_i}) = g(v) + log(\sum_{i=1}^{c} e^{f_i(x)}) \qquad (10)$$

So we can use $g(v)$ from the feature space, and $log(\sum_{i=1}^{c} e^{f_i(x)})$ from the output space to characterize the probability of a sample appearing. The greater the probability $P(v)$, the more likely the

sample is to belong to the ID sample. When $g(v) = max_j(-log(\sum_{i \neq j} e^{w_i^t v + b_i}))$, that is equal to MSP (Hendrycks & Gimpel, 2016). When $g(v) = 0$, that is equal to Energy (Liu et al., 2020). When $g(v)$ represents the inverse of the norm of the residuals of the projection of $v$ onto the main subspace, that is equal to VIM (Wang et al., 2022). When $g(v)$ is quadratic, that is GEM (Morteza & Li, 2022). We can look at these methods under a unified framework.

For practice, if we think that the ID samples are more likely to be close to the set $D_k = \{v \in R^N | v_i \leq k, i = 1, ..., N\}$. $g(v)$ is used to penalize samples that do not belong to set D. The measure of penalty takes the $L1$ distance from $v$ to the set $D_k$. So we can design our own novel score as:

$$G(x) = \sum_i (min(v_i, k) - k) + log(\sum_i e^{f_i(x)}) \tag{11}$$

Ignore the constant term, and provide a balance coefficient $\alpha$ for the information from the feature space and the output space. We can use the score:

$$G(x) = log(\sum_i e^{f_i(x)}) + \alpha \sum_i min(v_i, k) \tag{12}$$

Similar to the method we proposed in equation 8, they both use the same $U$ and $V$, the difference is that the combination is changed from multiplication to addition.

A natural idea is to choose a balance coefficient $\alpha$ that makes the standard deviations of the two terms close. Since the choice of the balance coefficient is intuitive, we use equation 8 by default in the experimental part. But we will do experiments to explore the appropriate balance coefficient.

## 4 EXPERIMENT

In this section, we evaluate RG on a large-scale OOD detection benchmark with ImageNet-1k as an in-distribution dataset. We describe the experimental setup in Section 4.1 and demonstrate the superior performance of RG over existing approaches in Section 4.2, followed by extensive ablations and analyses that improve the understanding of our approach.

### 4.1 EXPERIMENTAL SETUP

**Dataset** We evaluate our method on the large-scale ImageNet benchmark proposed by Huang & Li (2021). ImageNet benchmark is not only rich in data sources, but also many categories. OOD detection for the ImageNet model is more challenging due to both a larger feature space (dim = 2048) as well as a larger label space (C = 1000). In particular, the large-scale evaluation can be relevant to real-world applications, where the deployed models often operate on images that have high resolution and contain many class labels. Moreover, as the number of feature dimensions increases, noisy signals may increase accordingly, which can make OOD detection more challenging. We evaluate on four OOD test datasets, which are from subsets of iNaturalist (Van Horn et al., 2018), SUN (Xiao et al., 2010), Places (Zhou et al., 2017), and Textures (Cimpoi et al., 2014), with non-overlapping categories w.r.t. ImageNet-1k. The OOD datasets include various domains including fine-grained images, scene images, and textural images. The amount of OOD data is also very large, with the exception of Textures which has 5640 images, the other datasets have 10000 images each.

**Model and hyperparameters** We mianly use Google BiT-S models (Kolesnikov et al., 2020) pre-trained on ImageNet-1k with a ResNetv2-101 architecture (He et al., 2016). The BiT-S model is adopted not only for its excellent classification performance on ImageNet-1k but also for a better fair comparison with Gradnorm (Huang et al., 2021) methods. In Section 4.2. Additionally, we use clipping threshold 1 as the default and explore the effect of other clipping thresholds in Section 4.2. The temperature parameter T is set to be 1 unless specified otherwise, and we explore the effect of different temperatures in Section 4.2. We also report performance on another architecture, DenseNet121 (Huang et al., 2017). At test time, all images are resized to 480 × 480.

### 4.2 RESULTS AND ABLATION STUDIES

**Comparison with benchmark methods** The results for ImageNet evaluations are shown in 1, where our method(RG) demonstrates superior performance. We report OOD detection performance for

Table 1: **Main Results.** OOD detection performance comparison between RG and baselines. All methods utilize the standard ResNetv2-101 model trained on ImageNet. The classification model is trained on ID data only. All values are percentages.

| Method | iNaturalist | | SUN | | Places | | Textures | | Average | |
|---|---|---|---|---|---|---|---|---|---|---|
| | FPR95 | AUROC | FPR95 | AUROC | FPR95 | AUROC | FPR95 | AUROC | FPR95 | AUROC |
| MSP | 63.69 | 87.59 | 79.98 | 78.34 | 81.44 | 76.76 | 82.73 | 75.45 | 76.96 | 79.29 |
| KL-matching | 27.36 | 93.00 | 67.52 | 78.72 | 72.61 | 76.49 | 49.70 | 87.07 | 54.30 | 83.82 |
| Energy | 64.91 | 88.48 | 65.33 | 85.32 | 73.02 | 81.37 | 80.87 | 75.79 | 71.03 | 82.74 |
| ReAct | 49.97 | 89.80 | 65.30 | 87.40 | 73.12 | 85.34 | 80.82 | 70.53 | 67.30 | 83.27 |
| Mahalanobis | 96.34 | 46.33 | 88.43 | 65.20 | 89.75 | 64.46 | 52.23 | 72.10 | 81.69 | 62.02 |
| MOS | **9.28** | **98.15** | 40.63 | 92.01 | 49.54 | 89.06 | 60.43 | 81.23 | 39.97 | 90.11 |
| ODIN | 62.69 | 89.36 | 71.67 | 83.92 | 76.27 | 80.67 | 81.31 | 76.30 | 72.99 | 82.56 |
| Conor Igoe | 45.64 | 91.59 | 41.67 | 91.04 | 56.25 | 87.20 | 68.67 | 81.67 | 53.06 | 87.88 |
| Gradnorm | 50.03 | 90.33 | 46.48 | 89.03 | 60.86 | 84.82 | 61.42 | 81.07 | 54.70 | 86.31 |
| OURS | 31.38 | 94.67 | **35.36** | **92.53** | **49.06** | **89.19** | **31.80** | **92.30** | **36.90** | **92.17** |

each OOD test dataset, as well as the average over the four datasets. For a fair comparison, all the methods use the same pre-trained backbone, without regularizing with auxiliary outlier data. Since our method is inspired by Gradnorm(Huang et al., 2021), the settings of the method compared in that article are also the same as it, Such as MSP(Hendrycks & Gimpel, 2016), ODIN (Liang et al., 2017), Mahalanobis(Lee et al., 2018), as well as Energy(Liu et al., 2020). We also compared KL matching(Hendrycks et al., 2019) and the methods MOS (Huang & Li, 2021) which use the same pre-trained model on the same dataset. And for Conor Igoe (Igoe et al., 2022), we use the $L1$ norm of feature and the Energy. In addition, we also compare the ReAct(Sun et al., 2021), which has the same clipping threshold 1, and uses the Energy score after clipping as the score. We reproduce ReAct. Other methods have been reproduced on mos (Huang & Li, 2021) or Conor Igoe (Igoe et al., 2022), so we reported the result from them. For a fair comparison, we primarily compare with methods utilizing a pre-trained discriminative network without regularizing with auxiliary outlier data.

RG outperforms the best gradient-based baseline Gradnorm by 5.86% in AUROC. RG also outperforms a competitive feature-based method, Mahalanobis, by 44.79% in FPR95. RG also outperforms the method ReAct by 30.4% FPR95. Compared with the current group-based OOD detection method MOS, RG has promoted 2.06%. RG is stable with relatively small differences on four OOD datasets. Besides, OOD detection can be achieved in one forward pass without the need for another back pass like GradNorm. Our method is computationally small and requires no additional storage space, almost the same as MSP or energy methods.

Table 2: Ablation on U and V. OOD detection performance by different U and V. All methods utilize the standard ResNetv2-101 model trained on ImageNet. The classification model is trained on ID data only. All values are percentages. U and V are combined by multiplication. The meaning of the first line is to use the U and V of GradNorm (GN). The meaning of the Second line is to use the V of GradNorm and U of Ours. The meaning of the fourth line is to only use the U of Ours.

| Method | | iNaturalist | | SUN | | Places | | Textures | | Average | |
|---|---|---|---|---|---|---|---|---|---|---|---|
| U | V | FPR95 | AUROC | FPR95 | AUROC | FPR95 | AUROC | FPR95 | AUROC | FPR95 | AUROC |
| GN | GN | 50.03 | 90.33 | 46.48 | 89.03 | 60.86 | 84.82 | 61.42 | 81.07 | 54.70 | 86.31 |
| Ours | GN | 45.64 | 91.59 | 41.67 | 91.04 | 56.25 | 87.20 | 68.67 | 81.67 | 53.06 | 87.88 |
| GN | Ours | 39.15 | 93.20 | 41.85 | 89.89 | 55.55 | 86.00 | 34.52 | **93.08** | 42.77 | 90.54 |
| Ours | 1 | 64.91 | 88.48 | 65.33 | 85.32 | 73.02 | 81.37 | 80.87 | 75.79 | 71.03 | 82.74 |
| 1 | Ours | 73.18 | 76.63 | 62.15 | 76.86 | 75.54 | 70.71 | 42.93 | 87.73 | 63.45 | 77.98 |
| Ours | Ours | **31.38** | **94.67** | **35.36** | **92.53** | **49.06** | **89.19** | **31.80** | 92.30 | **36.90** | **92.17** |

**Ablation on U and V.** We conduct experiments on the results of $U$ and $V$ alone and using $U$ and $V$ in combination. There are two directions of ablation experiments: ablation to separate feature space and output space, and ablation to Gradnorm. As we described in Section 3, the combined use of information from feature space and output space can achieve performance that surpasses either of them in OOD detection. On the other hand, we also noticed that the information from the output space plays a leading role in OOD detection, which can also reflect the effectiveness of the previous methods such as Energy that depend on the output space.

Compared to Gradnorm, we have an increase in replacing $V$ with energy alone. Replacing $U$ alone with the cropped 1-norm also improves. This also shows that our $U$ and $V$ are suitable for OOD detection based on the BiT network.

Table 3: The effect of the clipping thresholds. OOD detection based our method. Use default temperature. All values are percentages.

| k | iNaturalist | | SUN | | Places | | Textures | | Average | |
|---|---|---|---|---|---|---|---|---|---|---|
| | FPR95 | AUROC | FPR95 | AUROC | FPR95 | AUROC | FPR95 | AUROC | FPR95 | AUROC |
| 0.1 | 48.04 | 91.90 | 47.14 | 89.58 | 58.49 | 85.39 | 35.76 | 90.47 | 47.36 | 86.84 |
| 0.3 | 33.68 | 94.31 | 36.24 | 91.86 | 49.69 | 88.12 | 26.51 | 93.11 | 36.53 | 91.85 |
| 0.5 | **30.44** | 94.78 | **34.69** | 92.38 | **48.41** | 88.89 | **25.66** | **93.43** | **34.80** | 92.37 |
| 0.7 | 30.67 | **94.82** | 34.98 | **92.52** | 48.86 | 89.14 | 27.60 | 93.15 | 35.53 | **92.41** |
| 1 | 31.38 | 94.67 | 35.36 | 92.53 | 49.06 | **89.19** | 31.80 | 92.30 | 36.90 | 92.17 |
| 2 | 35.59 | 93.78 | 37.54 | 92.17 | 51.84 | 88.71 | 42.55 | 88.80 | 41.88 | 90.87 |
| 3 | 39.20 | 93.01 | 39.18 | 91.77 | 53.73 | 88.17 | 50.12 | 86.00 | 45.56 | 89.74 |
| 5 | 43.30 | 92.10 | 40.83 | 91.29 | 55.44 | 87.55 | 56.00 | 83.12 | 48.89 | 88.52 |
| $\infty$ | 45.64 | 91.59 | 41.67 | 91.04 | 56.25 | 87.20 | 68.67 | 81.67 | 53.06 | 87.88 |

**The effect of the clipping thresholds** We evaluate our method RG with different clipping thresholds k from k = 0.1 to k = 5. As shown in 3, k = 0.5 or 0.7 is optimal, while either increasing or decreasing the clip threshold will degrade the performance. The appropriate clipping threshold is related to the distribution of ID samples in the feature space.

If the clipping threshold is relatively large, OOD samples with few strong features will be identified as ID samples. If the clipping thresholds are relatively small, the information loss of the samples in the feature space will be serious, which will lead to ID samples being identified as OOD samples. As the clipping threshold increases, the results converge to the unclipped result, which is the result of the last row. When the clipping is small, the performance will be worse than no clipping. This shows that a suitable clipping threshold is beneficial to filter out those OOD samples that make overconfident predictions because of extremely strong features.

Table 4: The effect of temperature. OOD detection based our method. Use default clipping threshold. All values are percentages.

| T | iNaturalist | | SUN | | Places | | Textures | | Average | |
|---|---|---|---|---|---|---|---|---|---|---|
| | FPR95 | AUROC | FPR95 | AUROC | FPR95 | AUROC | FPR95 | AUROC | FPR95 | AUROC |
| 0.5 | 32.84 | **95.56** | 41.26 | 91.27 | 54.55 | 87.80 | 39.50 | 90.60 | 42.04 | 91.31 |
| 1 | **31.38** | 94.67 | **35.36** | 92.53 | **49.06** | 89.19 | **31.80** | 92.30 | **36.90** | **92.17** |
| 2 | 62.63 | 85.40 | 52.53 | 85.09 | 66.97 | 80.55 | 37.94 | 90.98 | 55.02 | 85.51 |
| 4 | 72.13 | 77.74 | 60.98 | 78.01 | 74.57 | 72.06 | 42.45 | 88.06 | 63.03 | 78.97 |
| 8 | 73.04 | 76.83 | 62.01 | 77.07 | 75.46 | 70.96 | 42.87 | 87.77 | 63.35 | 78.16 |
| 16 | 73.14 | 76.70 | 62.07 | 76.90 | 75.51 | 70.77 | 42.93 | 87.74 | 63.41 | 78.03 |
| 32 | 73.18 | 76.67 | 62.14 | 76.87 | 75.54 | 70.72 | 42.93 | 87.73 | 63.45 | 77.80 |
| 64 | 73.18 | 76.66 | 62.15 | 76.86 | 75.54 | 70.71 | 42.93 | 87.73 | 63.55 | 77.99 |

**The effect of temperature.** We evaluate our method RG with different temperatures T. As shown in 4, T = 1 is optimal, while either increasing or decreasing the temperature will degrade the performance. This can be explained as the temperature balances the information in the feature space and the input space. The higher the temperature, the stronger the dependence on the feature space, and the lower the temperature, the stronger the dependence on the output space. As the temperature increases, the results converge to the penultimate row in 2, which is a result that depends entirely on the feature space.

**U plus V exploration.** As described in Section 3.3, OOD detection can also be performed using an additive combination of U and V. we test the OOD detection performance based on equation 12. Then the key to the experiment is the choice of the hyperparameter $\alpha$, this is a parameter that balances the effect of feature space and output space on OOD. A heuristic is to make the standard deviation of the two terms close, which we denote by $\alpha_s$. $\alpha_s$ need to be obtained from the training data of the network. To calculate it quickly, we randomly select 1k values from the training set to calculate. The experimental results are shown in Table 5. All of the average AUROC are better than the MOS which achieves an average AUROC of 0.901. On datasets where the output space benefits OOD detection more, the optimal alpha is slightly less than 1. On datasets where the feature space benefit OOD detection more, the optimal alpha is slightly greater than 1.

Table 5: The effect of balance factors. OOD detection based equation 12. All values are percentages.

| | iNaturalist | | SUN | | Places | | Textures | | Average | |
|---|---|---|---|---|---|---|---|---|---|---|
| $\alpha$ | FPR95 | AUROC | FPR95 | AUROC | FPR95 | AUROC | FPR95 | AUROC | FPR95 | AUROC |
| $0.3\alpha_s$ | 31.40 | 94.88 | 36.08 | 91.78 | 49.68 | 87.81 | 40.92 | 87.97 | 39.52 | 90.61 |
| $0.4\alpha_s$ | **29.20** | 94.95 | **34.82** | 92.41 | **48.46** | 88.73 | 36.17 | 89.89 | 37.16 | 91.50 |
| $0.5\alpha_s$ | 29.66 | **95.03** | 34.83 | **92.64** | 48.54 | 89.20 | 32.70 | 91.18 | 36.43 | 92.01 |
| $0.6\alpha_s$ | 30.27 | 94.88 | 35.03 | 92.60 | 49.03 | **89.32** | 31.12 | 92.04 | **36.36** | **92.21** |
| $0.8\alpha_s$ | 35.03 | 94.20 | 37.20 | 92.05 | 51.78 | 88.85 | 30.23 | 92.97 | 38.56 | 92.02 |
| $\alpha_s$ | 39.16 | 93.19 | 39.39 | 91.20 | 54.31 | 87.91 | **30.20** | 93.32 | 40.77 | 91.39 |
| $1.2\alpha_s$ | 42.91 | 92.32 | 41.62 | 90.25 | 56.50 | 86.81 | 30.76 | **93.37** | 42.95 | 90.69 |

Table 6: OOD detection performance comparison on a different architecture, DenseNet-121. Model is trained on ImageNet-1k as the ID dataset. All methods are post hoc and can be directly used for pre-trained models

| | iNaturalist | | SUN | | Places | | Textures | | Average | |
|---|---|---|---|---|---|---|---|---|---|---|
| Method | FPR95 | AUROC | FPR95 | AUROC | FPR95 | AUROC | FPR95 | AUROC | FPR95 | AUROC |
| MSP | 48.55 | 89.16 | 69.39 | 80.46 | 71.42 | 80.11 | 68.51 | 78.69 | 64.47 | 82.11 |
| Energy | 36.39 | 93.29 | 54.91 | 86.53 | 59.98 | 84.29 | 53.87 | 85.07 | 51.29 | 87.30 |
| ReAct | 44.25 | 91.14 | 63.48 | 85.48 | 68.52 | 82.93 | 69.15 | 78.17 | 61.35 | 84.43 |
| Mahalanobis | 97.36 | 42.24 | 98.24 | 41.17 | 97.32 | 47.27 | 62.78 | 56.53 | 88.93 | 46.80 |
| ODIN | 37.00 | 93.29 | 57.30 | 86.12 | 61.91 | 84.14 | 56.49 | 84.62 | 53.18 | 87.04 |
| Gradnorm | 23.87 | 93.97 | 43.04 | 87.79 | 53.92 | 83.04 | 43.16 | 87.48 | 41.00 | 88.07 |
| Conor Igoe | 19.05 | 95.93 | 37.46 | 90.73 | 47.65 | 87.37 | 37.59 | 89.08 | 35.44 | 90.78 |
| OURS | 14.91 | 96.93 | 34.80 | 91.45 | 45.59 | 88.32 | 27.52 | 92.77 | 30.71 | 92.36 |
| OURS(clips=0.7) | 14.48 | 97.01 | 34.48 | 91.36 | 45.14 | 88.33 | 25.90 | 93.01 | 30.00 | 92.43 |
| OURS(clips=0.5) | 14.46 | 97.01 | 35.16 | 91.17 | 45.28 | 88.24 | 25.53 | 93.07 | 30.11 | 92.37 |

**RG is effective on alternative neural network architecture.** We evaluate RG on a different architecture DenseNet-121 and report performance in Table 6. For a fair comparison, we reproduction ReActSun et al. (2021) and Connor Igoe Igoe et al. (2022), and other numbers were reported in Huang et al. (2021). RG is consistently effective, outperforming our source of inspiration Gradnorm by 10.29% in FPR95 and 4.29% in AUROC. If we use the optimal clipping threshold 0.5 shown in 3, FPR95 can drop 0.6% compared to the default clipping threshold 1. This shows that the appropriate clipping threshold is related in different network structures. Additionally, we also compare with state-of-the-art nonparametric feature space methods KNN (Sun et al., 2022). Because that method requires relatively high storage space, We decided to compare on the same ResNet-50 model trained on ImageNet. The results are shown in the table 7. we report KNN-based results from Sun et al. (2022). This also shows that our method is effective on another pre-trained model.

## 5    RELATED WORKS

**OOD detection by Output-based Methods** The earliest OOD detection method is based on MSP, which uses the maximum softmax probability as the indicator score of ID data (Hendrycks & Gimpel, 2016). The researchers' interest then turned to study OOD scores in the output space (Sastry & Oore, 2020; Dong et al., 2022). ODIN (Liang et al., 2017; Hsu et al., 2020) is an output-based method that uses temperature scaling and input perturbation to increase the separability of ID and OOD. After that, researchers' interest shifted from softmax space to logit space. (Liu et al., 2020) proposed using an energy score for OOD detection, which enjoys theoretical interpretation from a likelihood perspective (Morteza & Li, 2022). JointEnergy score (Wang et al., 2021) is then proposed to perform OOD detection for multi-label classification networks. Some recent studies have shown that one of the reasons for the overconfidence of OOD is the abnormally high activation of a few neurons, so appropriate inhibition of activated neurons is beneficial for OOD detection, which is ReAct (Sun et al., 2021). Then (Sun & Li, 2022) proposes a weight sparsification-based OOD detection framework termed DICE. These methods have the advantage of being easy to use without modifying the training procedure and objective.

**OOD detection by Feature-based Methods.** OOD detection based on feature space is often based on the assumption that after modeling the density function of ID data, OOD data is often in low-density regions, or OOD data is far from the center of ID samples Xiao et al. (2010); Zong et al.

Table 7: OOD detection performance comparison with KNN-based OOD detection. All methods utilize the ResNet-50 model trained on ImageNet. The classification model is trained on ID data only. All values are percentages.

| Method | iNaturalist | | SUN | | Places | | Textures | | Average | |
|---|---|---|---|---|---|---|---|---|---|---|
| | FPR95 | AUROC | FPR95 | AUROC | FPR95 | AUROC | FPR95 | AUROC | FPR95 | AUROC |
| KNN | 59.00 | 86.47 | 68.82 | 80.72 | 76.28 | 75.76 | 11.77 | 97.07 | 53.97 | 85.01 |
| -With Contrastive Learning | 30.18 | 94.89 | 48.99 | 88.63 | 59.15 | 84.71 | 15.55 | 95.40 | 38.47 | 90.91 |
| OURS | 20.21 | 96.02 | 31.12 | 92.62 | 42.83 | 89.73 | 26.03 | 92.88 | 30.05 | 92.81 |
| OURS(clips=0.7) | 18.89 | 96.28 | 30.66 | 92.74 | 42.22 | 89.88 | 23.74 | 93.34 | 28.88 | 93.06 |
| OURS(clips=0.5) | 18.12 | 96.43 | 30.58 | 92.70 | 42.35 | 89.34 | 22.29 | 94.00 | 28.34 | 93.12 |

(2018); Ren et al. (2019; 2021); Ming et al. (2022). A simple density assumption is that the features follow a class-conditional Gaussian distribution (Lee et al., 2018). For more complex distributions, flow technology can be used (Zisselman & Tamar, 2020). Nonparametric methods for estimating density have also recently emerged like Cook et al. (2020). And also OOD detection using k-nearest-neighbor has shown good performance (Sun et al., 2022), but it is based on a large amount of known ID data and finding K-nearest neighbors in practical applications is not an easy task in terms of storage and computation.

**OOD detection by Fusing information from feature space and output space.** Recently, there have been some methods that directly or indirectly mix the feature space information and output space information for OOD detection. (Huang et al., 2021) uses the feature space and output space information implicitly. VIM (Wang et al., 2022) uses the reconstruction error in the feature space and the energy in the output space. Igoe et al. (2022) also use the information in the feature space and output space for OOD detection. But experiments show that our method is better in performance because we use a more appropriate distance in the feature space.

## 6 DISCUSSION

Gradient-based OOD detection can often be transformed into a combination of information about the feature space and output space, as in section 3.1, the process of the BP algorithm also indirectly reflects this. An important reason why our method outperforms the baseline is that it uses information from both feature space and output space. Using the proper fusion method makes sense for OOD detection. It is a good choice to fuse the information of energy and feature space.

Our method benefits a lot from the network structure. Because the BN layer (Ioffe & Szegedy, 2015) is widely used in image recognition, the training data does not shift too much from 0, this is an important reason why reactivation strategy can improve performance. The reactivation strategy in our method is similar to Sun et al. (2021), but we don't use the reactivated feature vector to get the energy score. In Section 3.3 we revisit the reactivation method based on distance. From this view, we penalized the deviation from the set $D$ to get $g(\cdot)$. VIM (Wang et al., 2022) considers the set $D$ to be a subspace for training data. OOD-based KNN (Sun et al., 2022) considers the set D to be a subset of feature vectors for training data. However, the choice of $g(\cdot)$ in Section 3.3 is only heuristic, a strict $g(\cdot)$ should also satisfy normalization equation: $\int \sum_{i=1}^{C} e^{g(v)+w_i^t v+b_i} dv = 1$. In the future, we might make some assumptions about $g(\cdot)$ and then learn the parameters of $g(\cdot)$ from the training data.

## 7 CONCLUSION

In this paper, we propose RG, a novel OOD uncertainty estimation approach utilizing information extracted from the feature space and output space. And we propose a framework for combining metrics in the feature space and energy in the output space for OOD detection. Experimental results show that our gradient-based method can improve the performance of OOD detection by up to 2.06% in AUROC, establishing superior performance. Extensive ablations provide further understanding of our approach. We believe that considering both feature space and output space information can improve the performance of OOD detection. At the same time, we hope our work draws attention to the strong promise of the OOD detection methods that combine information from feature space and output space.

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
