# OpenReview forum: "RG: OUT-OF-DISTRIBUTION DETECTION WITH REACTIVATE GRADNORM"
_ICLR.cc/2023/Conference — Submitted to ICLR 2023_

### Official Review · Reviewer_TTD6 · 2022-10-20

**Confidence:** 5
**Correctness:** 1
**Technical Novelty And Significance:** 3
**Empirical Novelty And Significance:** 3
**Recommendation:** 3

**Clarity, Quality, Novelty And Reproducibility:**

The presentation is terrible. The methodology, as well as the theoretical analysis, are not original. The proposed method is a simple combination of existing OOD methods.

**Strength And Weaknesses:**

Pros:

1. The experiments are thorough and the proposed methods achieved promising results.

2. The addition-based formulation is new to the OOD regime, but its effectiveness is not very surprising.

Cons:

1. This idea is not surprising to me and the novelty is limited. First, the relationship between the gradient norm and the output/feature norm has been thoroughly analyzed by [Huang et al. 2021]. The main idea of this work is to replace the U and V terms with two widely-used OOD scores, i.e. energy scores [1] and ReAct [2]. In other words, the proposed methods are simple combinations of existing OOD scores. In effect, an ensemble of two OOD measures is quite straightforward for performance improvement.
Moreover, the authors did not properly cite these works when adapting them to their own methods.

3. I didn't find much connection between the proposed method and the gradient norm except for the U-V terms. But the title of this work is 'OOD with Reactivated Gradnorm'.

3. The writing is really, really terrible, and I recommend the authors carefully proofread and polish it. And there are also some grammar mistakes and typos:

    (a) 'ImageNet benchmark is not only rich in data sources, but also many categories.' => 'in categories'

    (b) Both 'OOD' and 'ood' were used to represent the terminology 'out-of-distribution'

**Summary Of The Paper:**

This paper studies the gradient-based out-of-distribution (G-OOD) detection problem. By revisiting the current methods, the authors indicate that G-OOD algorithms can be transformed to G(x)=UV, where U is a term representing the output-based score and V indicates the feature-norm-based OOD score. To this end, this work adapts these terms by (1) setting U to the energy score; (2) setting V to the trimmed feature norm. The authors also explore an addition-based mechanism to fuse them. Experimental results show that the proposed methods largely improve the OOD detection performance.

**Summary Of The Review:**

The writing of this work is terrible, and far from the bar of a top-tier AI conference. The novelty is limited.

---

### Official Review · Reviewer_b6vP · 2022-10-21

**Confidence:** 3
**Correctness:** 3
**Technical Novelty And Significance:** 2
**Empirical Novelty And Significance:** 2
**Recommendation:** 5

**Clarity, Quality, Novelty And Reproducibility:**

Parts of the proposed approach could be explained further:

For KL(.) under Eq. 4, 1/C seems to be missing within the log.  The decomposition of G(x) into UV below Eq. 6 can be further explained.

Eq. 7: T is not explained until later.

Sec 3.2: why $e^{-v_i^2}$ for V?  Also, why $V = \sum(v_i,k)$ is an approximation, why not $\sum(1-e^{-v_i^2},k)$, the exponential form?  That is, why exponential form in the first place?

Above Eq 9, what is the purpose of $g(v)$?

The decomposition of G into UV is interesting.

Reproducibility seems reasonable.


**Strength And Weaknesses:**

Strengths:

1.  The decomposition of G, gradient, into U and V is interesting

2.  Their results indicate that their approach compares favorably against existing approaches.

Weaknesses:

1.  Parts of the proposed approach could be explained further (details in the next section)

2.  What are the principles/properties for choosing U and V besides outputs and features?  The proposed choices seem ad hoc.

**Summary Of The Paper:**

The authors propose, Reactivate Gradnorm (RG)  in feature space as well as output space for OOD.  Specifically, they multiply the 1-norm of the features and log of the exponential sum of the output.   Their motivation is from a gradient-based method.   For a trained network, if the target is uniform distribution, an in-distribution (ID) instance x will have a large gradient G(x).  If G(x) < c, some threshold, the instance is rejected as OOD .   G(x) can be decomposed into UV, where V depends on the features and U depends on the output.  They propose using an energy based score for U and clipped 1-norm for V.  They further propose a variant that uses U+V, instead of UV.

The compared their approach (RG, the UV version) with 9 existing approaches with 4 datasets.  Their results indicate that their approach compare favorably.  For ablation study, they varied the choice of U and V and found their proposed U and V are generally more desirable.  They also perform sensitivity analysis on parameters T (temperature), k (clipping threshold).

**Summary Of The Review:**

The decomposition of G into UV is interesting.  Their results indicate their approach compares favorably.  However, the proposed choices of U and V seem ad hoc.  Also, parts of the approach could be explained further.

---

### Official Review · Reviewer_7gKP · 2022-10-23

**Confidence:** 3
**Correctness:** 2
**Technical Novelty And Significance:** 2
**Empirical Novelty And Significance:** 2
**Recommendation:** 5

**Clarity, Quality, Novelty And Reproducibility:**

The quality and the clarity of writing are satisfactory. The Novelty may be limited to some extent. Moreover, I did not check the reproducibility.

**Strength And Weaknesses:**

- The authors adopt the gradient information in discerning ID and OOD data. I also believe it is a very important line of research in OOD detection. However, to me, it is not crystal clear *why the proposed method can be superior over previous works in using gradient information*. Especially, the authors claim that the existing works still suffer from the problem of overconfidence. So, two natural questions are 1) why previous works are suffer from overconfident issue and 2) why the proposed method can mitigate such overconfidence. I think the related discussion can be critical, but I am afraid that I cannot find much useful information throughout the paper.

- *The novelty of this paper is limited*. The authors followed previous works [1,2] in calculating the gradient norm of the last layer without back propagation (Eq. 6), and their choices of U and V also follow the previous works in energy scoring [3] and ReAct [4]. So, I think the authors directly combine the several advanced works in previous studies (actually, there are advanced works try to combine different scoring strategies for the improved detection capability of the model [5]), and I am not sure what makes such combination indispensable (for example, one can also change the ReAct by the L2 norm of logit features, so why the authors prefer the ReAct over other advanced scoring strategies).

[1] Conor Igoe, et al. How Useful are Gradients for OOD detection Really? 2022.

[2] Rui Huang, et al. One the Importance of Gradients for Detecting Distribution Shifts in the Wild. NeurIPS, 2022.

[3] Weitang Liu, et al. Energy-based Out-of-distribution Detection. NeurIPS, 2020.

[4] Yiyou Sun, et al. ReAct: Out-of-distribution Detection with Rectified Activations. NeurIPS, 2021.

[5] Hihoon Tack, et al. CSI: Novelty Detection via Contrastive Learning on DIstributionally Shifted Instances. NeurIPS, 2020.

- The paper is motivated by the GradNorm, but *the finally adopted method does not involve much about the gradient information*. So, I am not sure if the addition-based combination can benefit from the gradient information of the model. It will be great if the authors could formally connect the addition-based combination and the GradNorm in math language.

- The authors conduct experiments on ImageNet benchmark, which is a challenging setup with large semantic space and very complex data features. However, *the comparison with some advanced methods (e.g., [6,7]) and the experiments on Hard OOD detection (e.g., CIFAR-10 vs. CIFAR-100, cf., [6]) are missing*. To fully justify the effectiveness of the proposal, I am afraid the authors should conduct more experiments.

[6] Yiyou Sun, et al., Out-of-distribution Detection with Deep Nearest Neighbors. ICML, 2022.

[7] Xuefeng Du, et al. VOS: Learning What You Don't Know by Virtual Outlier Synthesis. ICLR, 2022.



**Summary Of The Paper:**

Detecting OOD data is critical to build reliable machine learning systems, where the models should make reliable predictions for ID data meanwhile detecting OOD data without further predictions. It motivates the recent studies in OOD detection, which has attracted significant attentions recently. The authors propose a novel method named Reactivate Gradnorm, which exploits the norm of clipped feature vector and the energy in the output space for OOD detection. The authors conducted experiments on ImageNet, and the results demonstrate their superiority over the state-of-the-art approaches.

**Summary Of The Review:**

Using gradient information in OOD detection is an interesting direction. However, to improve the quality of the paper, I think the authors can address my concerns.

---

> ### Author Response · Authors · 2022-11-08
> **We would like to express our gratitude to you for taking the valuable time to read our manuscript. At the same time, we sincerely thank you for pointing out the shortcomings in the manuscript. Your valuable comments help us improve the paper. We revise the manuscript using the comments as guidelines and supply missing contents, improve readability  to better present our work. We try to address each comment as satisfactorily as possible and hope to remove all drawbacks pointed out by reviewers.**
>
> First we will restate our method from the gradnorm perspective：
> Gradient-based OOD detection methods usually adopt a design loss function and exploit the norm of the gradient of its last layer. This method must be able to decompose G into a combination of output layer information U and output layer information V. This is because：
> The parameter of the last layer of the neural network is w, then the input of the last layer is v, then the output of the neural network is f = Wv. The loss function is L. Then $G(x) = |\frac{\partial L}{\partial W}| = |\frac{\partial L}{\partial f}v^t| =   |\frac{\partial L}{\partial f}|| v^t|$. Gradnorm uses uniform labels and outputs to calculate KL divergence. Reactivate Gradnorm  uses the output energy and 0 to calculate the mean square loss， then $L = 0.5(log \sum e^{f_i}-0)^2$,$G(x) = (log \sum e^{f_i}) \sum |v_i| $. Then apply the gradient clipping method, which is a commonly used method to avoid exploding gradients in neural network training. We clip gradients over 1 to 1. In theory, we should apply the gradient clipping operator to $\frac{\partial L}{\partial f} v^t$. But we only apply gradient clipping to the latter term, and then we get $G(x) = (log \sum e^{f_i}) \sum max(v_i,1)$, because this renders the experiment better.
>
> 1.Now to answer the first question, why does the previous method have the problem of overconfidence prediction and why our method can solve this problem.
> (1). One reason why our method can overcome overconfidence predictions is that our method makes better use of gradient information than Gradnorm. Compared with calculating KL divergence for uniform labels, calculating the mean square error between energy and 0 can avoid identifying fuzzy samples in ID samples as OOD samples，Because such samples have high energy, but predicted probabilities are close to uniform labels.
>
> (2). Another reason comes from the inherent flaws of the gradnorm method. The reason why the gradient information can be used for OOD detection is that the parameter update of the neural network tends to 0 when the pre-training model is trained, the parameter update is not equal to the product of the negative gradient and the learning rate. It is also affected by factors such as gradient clipping, weight decay, and momentum. Compared with Gradnorm, our method takes into account the effect of gradient clipping more.
>
> 2.From a gradient perspective, we consider gradient clipping operation commonly used in optimizers, which is why we prefer ReAct operations. More generally, we can also consider the weight decay operation commonly used in optimizers. This will cause us to ignore values close to 0 in OOD detection in addition to clipping larger values. As far as innovation is concerned, we think our work actually gives us a direction to use gradient information to find good OOD detection. That is, to find the appropriate output layer information U instead of finding the appropriate loss function L. Since $U = \frac{\partial L}{\partial f}$, If the optimal L can be represented by a composite of elementary functions, then U must also be represented by a composite of elementary functions,  not vice versa. That is to say, when designing U, we are actually implicitly designing L in a larger function class. We think future approaches to OOD detection based on the norm of gradients should focus on designing U rather than L, although L is more interpretable.
>
> 3.We just rephrased our method more in terms of gradients, our method is actually obtained by computing the gradient of the mean squared loss between energy and 0, and then applying the gradient clipping.  It is easy to convert between U+V and UV.Because U+V is equivalent to e^{U}e^{V}$. The reason why we proposed the addition-based score is that  we can look at MSP, Energy, GEM, VIM and our own method in (12) under a unified framework.
>
> 4.The results of the gradient-based OOD method on the Cifar database are not good, such as cifar as the ID sample and Mnist as the OOD sample. Methods like Gradnorm or Energy are not even as effective as the simplest baseline MSP [1].  I think our assessment on imagmet is enough，becasuse ImageNet is a challenging setup with large semantic space and very complex data features. In addition, we compared [2] in Table 7. Due to the computational and storage resources required by the KNN method, we do not reproduce their results, but instead report our results on the same pretrained models they use. As for VOS [3], the reason why we do not compare this method is that it requires an additional training process for the model. This approach fine-tunes the pretrained model. In contrast, our method is simpler and easier to implement.
>
> [1] Conor Igoe, et al. How Useful are Gradients for OOD detection Really? 2022.
>
> [2]Yiyou Sun, et al., Out-of-distribution Detection with Deep Nearest Neighbors. ICML, 2022.
>
> [3]Xuefeng Du, et al. VOS: Learning What You Don't Know by Virtual Outlier Synthesis. ICLR, 2022.

---

> > ### Comment · Reviewer_7gKP · 2022-11-08
> > **Thanks for the authors' feedbacks**
> >
> > Overall, I would like to keep my score as 5, after reading authors’ feedback and other reviewers’ comments. Here I would like to list  two reasons.
> >
> > > The authors refuse to report the results on CIFAR benchmarks.
> >
> > The authors state that the Gradnorm and Energy are not good on the CIFAR benchmarks, and the proposed method inherents the drawbacks of these methods. However, it is contradictory to authors' previous claims that the proposed method can improve Gradnorm, and the energy can overcome the overconfidence issue. Therefore, I do not think that the authors' claims in the paper is well supported.
> >
> > Further, the authors claim that the assessment on ImageNet is enough, because ImageNet is more challenging. So, why the proposed method does not work well even on a much simple setup? I am afraid that the proposed method is not general enough, and the usage scenario of this method is limited.
> >
> > Besides, there are a lot of challenging experimental setup for the CIFAR benchmarks, such as the hard OOD detection, just as I mentioned. Sadly, the authors refuse to report the related results, and I am concern that the proposed method is not as effective as authors' claims.
> >
> > > The authors cannot explain the reasons for their design choices of RG.
> >
> > I believe the novelty of this paper is limited, which is supported by the comments of other reviewers. Sadly, the authors do not try to provide a strong feedback about why RG is novel.
> >
> > Further, the authors mainly have three points of modifications over GradNorm, i.e., gradient clipping, MSE between energy and 0, and addition-based score. However, the question in why these modifications can be helpful is not well discussed, just as commented by Reviewer b6vP.
> >
> > I hope the authors could try to address the following issues, which I think they are helpful to improve the paper's quality.
> >
> > 1. For gradient clipping, the authors claim that it is adopted since "it is a commonly used method to avoid exploding gradients in neural network training". It seems that the authors mainly focus on the post-hoc approaches, involving no model training. So, why a technique for model training can be used to address the over-confidence issue in OOD detection.
> >
> > 2. The authors claim that "In theory, we should apply the gradient clipping operator to $\frac{dL}{df}v^t$". So, where is the theoretical justification. I hope the authors could provide a formal theorem about this point, justifying why such a methodology can overcome the over-confidence issue and why such a methodology can improve OOD detection.
> >
> > 3. For the energy scoring, the authors claim that using the energy scoring can address the overconfidence issue. I agree with this point of authors' claim. However, I do not understand what is the superiority of RG over directly using free energy scoring. Why introducing gradient information can improve free energy scoring? Why introducing gradient information can address the overconfidence issue?
> >
> > 4. I cannot understand why the addition-based score can lead to the improved results over multiplicative score. If the addition-based score do not lead to improved results over the multiplicative counterpart, why the authors prefer using addition-based scoring?
> >
> > 6. Further, the authors claim that "the reason why we proposed the addition-based score is that we can look at MSP, Energy, GEM, VIM and our own method in (12) under a unified framework". Could the authors explain exactly what is the unified framework mentioned here, and what kind of knowledge we can achieve from such a unified framework in devising better methods  for OOD detection?

---

> > > ### Author Response · Authors · 2022-11-17
> > > **Thanks for your feedbacks.**
> > >
> > > After several days of thinking, I have to admit that I have a limited ability to make strict answers to theoretical things. When I was doing this experiment, I didn't have a clear theory to guide me,  I was just looking for some simple and effective OOD detection scores.
> > >
> > > There are only some vague things guiding me to make appropriate corrections.
> > > Such as:
> > > Compared with MSP, energy score can utilize more information on non-maximum prediction categories
> > > Using the gradient of loss rather than the loss itself may be more beneficial to OOD detection
> > > Suppressing large values can reduce the impact of outliers
> > > The use of the addition rule is similar to VIM, and adjusting the balance coefficient is also likely to improve the performance.
> > >
> > > In other words, our motivation comes from intuition, and then further experiments have improved the OOD detection performance. It is difficult to explain it strictly in theory.

---

### Official Review · Reviewer_diYs · 2022-10-23

**Confidence:** 3
**Correctness:** 3
**Technical Novelty And Significance:** 2
**Empirical Novelty And Significance:** 2
**Recommendation:** 3

**Clarity, Quality, Novelty And Reproducibility:**

The novelty and Quality are fair. This work contributes some new ideas. It has minor technical flaws and some typos. The errors are fixable. The clarity is poor. The content should be carefully reorganized.

**Strength And Weaknesses:**

Strength:

- This work proposes a new idea to use the decomposition of gradient-based OOD detection.
- The proposed method is simple and easy to implement.
- The empirical result in Table 7 is good.

Weakness:

- The proposed method is not well-motivated.
- The novelty of the proposed method is unclear.
- The content is poorly organized.
- This work lacks a hyperparameter selection method based on ID data.


**Summary Of The Paper:**

The authors revisit gradient-based OOD detection from the perspective of backpropagation and extend the decomposition G(x)=UV of GradNorm to more loss functions. Here G is a gradient-based detection score, V is the feature norm and U represents output information. According to this decomposition, the authors suggest exploiting suitable U and V for OOD detection. They take U as the energy-based score and derive a V measurement by assuming OOD features follow a standard Gaussian distribution. Experiments on four ImageNet benchmarks demonstrate the effect of the proposed method and discuss the impact of hyperparameters.

**Summary Of The Review:**

I list my main questions in this section.

1. In the abstract, the motivation is 'However, existing works still suffer from the problem of overconfidence'. Why the proposed score in (8) can overcome the problem? Could you provide more analysis and comparisons to GradNorm and other OOD detection scores?
2. In Section 3.1, does "ground truth" mean ground truth for classification or ground truth for OOD detection? Can you point out a loss function that corresponds to your proposed score in (8)?
3. Can we understand the proposed score as an enhancement for the energy-based score? Why did you name it "Reactivate Gradnorm"? Is it because the proposed score follows the decomposition G(x)=UV?
4. What is the purpose of introducing Section 3.3?
5. The main result with ResNetv2-101 (Table 1) is not as good as the result with ResNet 50 (Table 7). Why use different pre-trained models in this section? In Table 7, is it unfair to compare Reactivate GradNorm to KNN? Would it be more convincing to compare Reactivate GradNorm with KNN+ReAct?
6. The OOD detection task in this work is a one-sample hypothesis testing problem, i.e., only ID data is accessible. Therefore, the hyperparameters in your score should be determined by the ID data. Table 3 and Table 4 only consider one ID data.

---

> ### Author Response · Authors · 2022-11-08
> **We would like to express our gratitude to you for taking the valuable time to read our manuscript. At the same time, we sincerely thank you for pointing out the shortcomings in the manuscript. Your valuable comments help us improve the paper. We revise the manuscript using the comments as guidelines and supply missing contents, improve readability  to better present our work. We try to address each comment as satisfactorily as possible and hope to remove all drawbacks pointed out by reviewers.**
>
> 1.
>
> (1). Gradient-based OOD detection methods usually adopt a design loss function and exploit the norm of the gradient of its last layer. This method must be able to decompose G into a combination of output layer information U and output layer information V. This is because：
> The parameter of the last layer of the neural network is w, then the input of the last layer is v, then the output of the neural network is f = Wv. The loss function is L. Then $G(x) = |\frac{\partial L}{\partial W}| = |\frac{\partial L}{\partial f}v^t| =   |\frac{\partial L}{\partial f}|| v^t|$. Gradnorm uses uniform labels and outputs to calculate KL divergence. Reactivate Gradnorm  uses the output energy and 0 to calculate the mean square loss， then $L = 0.5(log \sum e^{f_i}-0)^2$,$G(x) = (log \sum e^{f_i}) \sum |v_i| $. Then apply the gradient clipping method, which is a commonly used method to avoid exploding gradients in neural network training. We clip gradients over 1 to 1. In theory, we should apply the gradient clipping operator to $\frac{\partial L}{\partial f} v^t$. But we only apply gradient clipping to the latter term, and then we get $G(x) = (log \sum e^{f_i}) \sum max(v_i,1)$, because this renders the experiment better.
>
> (2). One reason why our method can overcome overconfidence predictions is that our method makes better use of gradient information than Gradnorm. Compared with calculating KL divergence for uniform labels, calculating the mean square error between energy and 0 can avoid identifying fuzzy samples in ID samples as OOD samples，Because such samples have high energy, but predicted probabilities are close to uniform labels.
>
> (3). Another reason comes from the inherent flaws of the gradnorm method. The reason why the gradient information can be used for OOD detection is that the parameter update of the neural network tends to 0, and when the pre-training model is trained, the parameter update is not equal to the product of the negative gradient and the learning rate. It is also affected by factors such as gradient clipping, weight decay, and momentum. Compared with Gradnorm, our method takes into account the effect of gradient clipping more.
>
> 2.In the first part of my reply, we pointed out the specific form of the loss function, that is to say, taking the mean square loss of energy and 0, and then applying gradient clipping to get our method. By the way I don't think writing out the analytic form of the loss function is of much benefit for finding the optimal OOD detection score. Because assuming that we can use the composition of elementary functions to design an optimal loss function $L$, then we must be able to use $U = |\frac{\partial L}{\partial f}|$ to get a composite U that can use the composition of elementary functions. Not otherwise. Designing U is actually implicitly designing L in a function class larger than U.
>
> 3.As can also be seen from our answer in the first part, our method is an improvement of the Gradnorm method. We use a different loss function and use a gradient clipping strategy. The striking similarity in form stems from the fact that when f = Wv, $G(x) = |\frac{\partial L}{\partial W}| $ must be this decomposition  $|\frac{\partial L}{\partial f}|| v^t|$, that is G(x) = UV. It can be understood as an enhancement for the energy-based score, because this actually uses the gradient of the mean square error of energy and 0. From a gradient clipping perspective, our method should probably be called "Clip Gradnorm". But since we only clip a part of the gradient, that is, the feature vector, it is called "Reactivate Gradnorm".  Another reason to use “Reactivate” is that we do not want the clipping operation to limit the applicability of our method. Because from the perspective of weight decay, the influence of small gradients must be excluded. That is to say, we cannot use the gradient directly, but perform the "Reactivate" operation on the gradient. We can use the clipping operation $g(\cdot)=max(\cdot,1)$, of course we can also use something like $g(\cdot)=1-e^{-(\cdot)}$, the latter will achieve an average AUROC of 0.91 in Table 1, still outpacing all other baselines. So clipping may not be the core, the core is reactivation.
>
> 4.We want to propose a decomposition method that is formally different from G(x)=UV, namely G(x) = U+V in (10). with the help of conditional probability, we can look at MSP, Energy, GEM, VIM and our own method in (12) under a unified framework. By the way, it is easy to convert between addition-based OOD detection scores and multiplication-based ones. Because $G(x) =U + V$ is equivalent to $G(x)= e^{U}e^{V}.

---

> > ### Author Response · Authors · 2022-11-08
> > **Supplement to the previous one**
> >
> > 5.The reason for using different pretrained models is that we want to show that our method is robust to different pretrained models.In Table 7, maybe is it unfair to compare Reactivate GradNorm to KNN, so we compair the KNN with contrastive learning (named ‘KNN+’ in their paper). The use of contrastive learning actually also plays a role in changing the distribution of data in the feature space which ReAct does the same. So we think the comparion between Reactivate GradNorm with KNN+contrastive learning is fair. I agree it would be more convincing to compare Reactivate GradNorm with KNN+ReAct, but we can’t reproduct the KNN because of our computer's memory size limit.
> >
> > 6.As shown in table 3 and table 4, our method is robust to the choice of hyperparameters. Just use the default clipping threshold of 1 and temperature coefficient of 1 to get good results. I can also provide a heuristic choice. For example, I can set T to the T of the softmax layer during model training, and set the clipping threshold k to the value of gradient clipping during model training. Of course, they are all 1 by default.

---

### Decision · Program_Chairs · 2023-01-20

**Decision:**

Reject

**Justification For Why Not Higher Score:**

NA

**Justification For Why Not Lower Score:**

NA

**Metareview: Summary, Strengths And Weaknesses:**

This paper studies out-of-distribution (OOD) detection, one of the most important problems in modern machine learning. The authors propose a new OOD detection score called ``Reactivate Gradnorm (RG)'', which exploits gradient information of a pre-trained neural network. Compared to feature-based and output-based OOD detection, gradient-based methods are relatively underexplored. Hence, this paper investigates an interesting direction that adds value to the field.

The key idea of the paper draws on GradNorm by Huang et al. 2021, which showed that GradNorm can be decomposed into two multiplicative terms: $G(x) = U\cdot V$. Here, U encapsulates information in the output space and V indicates the $L_1$ norm of the penultimate feature vector. This paper took the inspiration and modified the original method in two interesting ways:

* RG changed the loss function in calculating the gradient: from the KL divergence between softmax and uniform target to the MSE of energy
* RG applied rectified activations (ReAct) in the feature vector before calculating the L_1 norm $|v|$. The proposed method is easy to implement and achieves promising performance.

While the overall design and method are reasonable, multiple reviewers are concerned about the limited novelty (i.e. direct combination of energy score by Liu et al. 2020 and ReAct by Sun et al. 2021), and writing clarity. This led to a unanimous vote for rejection among four reviewers. Some review comments from b6vP and TTD6 were left unaddressed. The paper in its current form might not be ready yet for ICLR, and is rejected due to a lack of broad support from reviewers.

The AC would like to encourage the authors to incorporate the review comments in the revision, and perhaps target the next venue with changes made. I do like the idea and motivation of thinking about the improved loss function for calculating GradNorm. As reviewer b6vP suggested, it might also be worth investigating more deeply the principles and properties of designing U and V, which can further strengthen the work. I also strongly encourage the authors to add a comparison with KNN (trained with cross-entropy loss) and KNN+ReAct in the revision, which would strengthen the argument for the gradient-based score that takes into account both output and feature information. Lastly, one minor writing suggestion is to differentiate terminology activation clipping vs. gradient clipping. The latter often refers to clipping directly in the parameter space based on $\frac{\partial L}{\partial w}$, whereas the former is closer to what the authors are trying to do: $|\frac{\partial L}{\partial f}| \cdot |\text{clip}(v)|$.

I hope the authors are not too discouraged by the decision, but rather, use the comments as a source of inspiration for better work in the future :)